# Linearly constrained Gaussian processes

**Carl Jidling**
Department of Information Technology
Uppsala University, Sweden
carl.jidling@it.uu.se

**Niklas Wahlström**
Department of Information Technology
Uppsala University, Sweden
niklas.wahlstrom@it.uu.se

**Adrian Wills**
School of Engineering
University of Newcastle, Australia
adrian.wills@newcastle.edu.au

**Thomas B. Schön**
Department of Information Technology
Uppsala University, Sweden
thomas.schon@it.uu.se

## Abstract

We consider a modification of the covariance function in Gaussian processes to correctly account for known linear operator constraints. By modeling the target function as a transformation of an underlying function, the constraints are explicitly incorporated in the model such that they are guaranteed to be fulfilled by any sample drawn or prediction made. We also propose a constructive procedure for designing the transformation operator and illustrate the result on both simulated and real-data examples.

## 1 Introduction

Bayesian non-parametric modeling has had a profound impact in machine learning due, in no small part, to the flexibility of these model structures in combination with the ability to encode prior knowledge in a principled manner [6]. These properties have been exploited within the class of Bayesian non-parametric models known as Gaussian Processes (GPs), which have received significant research attention and have demonstrated utility across a very large range of real-world applications [16].

Abstracting from the myriad number of these applications, it has been observed that the efficacy of GPs modeling is often intimately dependent on the appropriate choice of mean and covariance functions, and the appropriate tuning of their associated hyper-parameters. Often, the most appropriate mean and covariance functions are connected to prior knowledge of the underlying problem. For example, [10] uses functional expectation constraints to consider the problem of gene-disease association, and [13] employs a multivariate generalized von Mises distribution to produce a GP-like regression that handles circular variable problems.

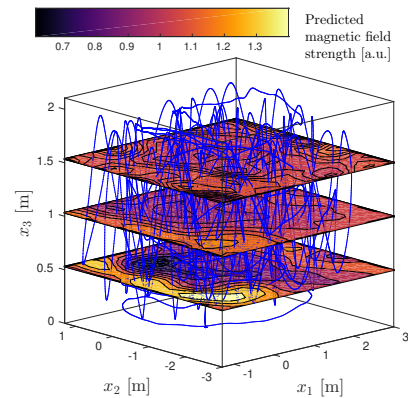

Figure 1: Predicted strength of a magnetic field at three heights, given measured data sampled from the trajectory shown (blue curve). The three components $(x_1, x_2, x_3)$ denote the Cartesian coordinates, where the $x_3$-coordinate is the height above the floor. The magnetic field is curl-free, which can be formulated in terms of three linear constraints. The method proposed in this paper can exploit these constraints to improve the predictions. See Section 5.2 for details.

At the same time, it is not always obvious how one might construct a GP model that obeys underlying principles, such as equilibrium conditions and conservation "laws". One straightforward approach to this problem is to add fictitious measurements that observe the constraints at a finite number of points of interest. This has the benefit of being relatively straightforward to implement, but has the sometimes significant drawback of increasing the problem dimension and at the same time not enforcing the constraints between the points of interest.

A different approach to constraining the GP model is to construct mean and covariance functions that obey the constraints. For example, curl and divergence free covariance functions are used in [22] to improve the accuracy for regression problems. The main benefit of this approach is that the problem dimension does not grow, and the constraints are enforced everywhere, not pointwise. However, it is not obvious how these approaches can be scaled for an arbitrary set of linear operator constraints.

The contribution of this paper is a new way to include constraints into multivariate GPs. In particular, we develop a method that transforms a given GP into a new, derived, one that satisfies the constraints. The procedure relies upon the fact that GPs are closed under linear operators, and we propose an algorithm capable of constructing the required transformation. We will demonstrate the utility of this new method on both simulated examples and on a real-world application, the latter in form of predicting the components of a magnetic field, as illustrated in Figure 1.

To make these ideas more concrete, we present a simple example that will serve as a focal point several times throughout the paper. To that end, assume that we have a two-dimensional function $\mathbf{f}(\mathbf{x}) : \mathbb{R}^2 \mapsto \mathbb{R}^2$ on which we put a GP prior $\mathbf{f}(\mathbf{x}) \sim \mathcal{GP}\left(\boldsymbol{\mu}(\mathbf{x}),\ K(\mathbf{x}, \mathbf{x}')\right)$. We further know that $\mathbf{f}(\mathbf{x})$ should obey the differential equation

$$\frac{\partial f_1}{\partial x_1} + \frac{\partial f_2}{\partial x_2} = 0. \tag{1}$$

In this paper we show how to modify $K(\mathbf{x}, \mathbf{x}')$ and $\boldsymbol{\mu}(\mathbf{x})$ such that any sample from the new GP is guaranteed to obey the constraints like (1), considering any kind of linear operator constraint.

## 2 Problem formulation

Assume that we are given a data set of $N$ observations $\{\mathbf{x}_k, \mathbf{y}_k\}_{k=1}^N$ where $\mathbf{x}_k$ denotes the input and $\mathbf{y}_k$ the output. Both the input and output are potentially vector-valued, where $\mathbf{x}_k \in \mathbb{R}^D$ and $\mathbf{y}_k \in \mathbb{R}^K$. We consider the regression problem where the data can be described by a non-parametric model $\mathbf{y}_k = \mathbf{f}(\mathbf{x}_k) + \mathbf{e}_k$, where $\mathbf{e}_k$ is zero-mean white noise representing the measurement uncertainty. In this work, we place a vector-valued GP prior on $\mathbf{f}$

$$\mathbf{f}(\mathbf{x}) \sim \mathcal{GP}\left(\boldsymbol{\mu}(\mathbf{x}),\ K(\mathbf{x}, \mathbf{x}')\right), \tag{2}$$

with the mean function and the covariance function

$$\boldsymbol{\mu}(\cdot) : \ \mathbb{R}^D \mapsto \mathbb{R}^K, \qquad\qquad K(\cdot, \cdot) : \ \mathbb{R}^D \times \mathbb{R}^D \mapsto \mathbb{R}^K \times \mathbb{R}^K. \tag{3}$$

Based on the data $\{\mathbf{x}_k, \mathbf{y}_k\}_{k=1}^N$, we would now like to find a posterior over the function $\mathbf{f}(\mathbf{x})$. In addition to the data, we know that the function $\mathbf{f}$ should fulfill certain constraints

$$\mathcal{F}_{\mathbf{x}}[\mathbf{f}] = \mathbf{0}, \tag{4}$$

where $\mathcal{F}_{\mathbf{x}}$ is an operator mapping the function $\mathbf{f}(\mathbf{x})$ to another function $\mathbf{g}(\mathbf{x})$ as $\mathcal{F}_{\mathbf{x}}[\mathbf{f}] = \mathbf{g}(\mathbf{x})$. We further require $\mathcal{F}_{\mathbf{x}}$ to be a linear operator meaning that $\mathcal{F}_{\mathbf{x}}\left[\lambda_1 \mathbf{f}_1 + \lambda_2 \mathbf{f}_2\right] = \lambda_1 \mathcal{F}_{\mathbf{x}}[\mathbf{f}_1] + \lambda_2 \mathcal{F}_{\mathbf{x}}[\mathbf{f}_2]$, where $\lambda_1, \lambda_2 \in \mathbb{R}$. The operator $\mathcal{F}_{\mathbf{x}}$ can for example be a linear transform $\mathcal{F}_{\mathbf{x}}[\mathbf{f}] = C\mathbf{f}(\mathbf{x})$ which together with the constraint (4) forces a certain linear combination of the outputs to be linearly dependent.

The operator $\mathcal{F}_{\mathbf{x}}$ could also include other linear operations on the function $\mathbf{f}(\mathbf{x})$. For example, we might know that the function $\mathbf{f}(\mathbf{x}) : \mathbb{R}^2 \to \mathbb{R}^2$ should obey a certain partial differential equation $\mathcal{F}_{\mathbf{x}}[\mathbf{f}] = \frac{\partial f_1}{\partial x_1} + \frac{\partial f_2}{\partial x_2}$. A few more linear operators are listed in Section 1 of the Supplementary material, including integration as one the most well-known.

The constraints (4) can either come from known physical laws or other prior knowledge of the process generating the data. Our objective is to encode these constraints in the mean and covariance functions (3) such that any sample from the corresponding GP prior (2) always obeys the constraint (4).

# 3 Building a constrained Gaussian process

## 3.1 Approach based on artificial observations

Just as Gaussian distributions are closed under linear transformations, so are GPs closed under linear operations (see Section 2 in the Supplementary material). This can be used for a straightforward way of embedding linear operator constraints of the form (4) into GP regression. The idea is to treat the constraints as noise-free artificial observations $\{\tilde{\mathbf{x}}_k, \tilde{\mathbf{y}}_k\}_{k=1}^{\tilde{N}}$ with $\tilde{\mathbf{y}}_k = \mathbf{0}$ for all $k = 1 \ldots \tilde{N}$. The regression is then performed on the model $\tilde{\mathbf{y}}_k = \mathscr{F}_{\tilde{\mathbf{x}}_k}[\mathbf{f}]$, where $\tilde{\mathbf{x}}_k$ are input points in the domain of interest. For example, one could let these artificial inputs $\tilde{\mathbf{x}}_k$ coincide with the points of prediction.

An advantage of this approach is that it allows constraints of the type (4) with a non-zero right hand side. Furthermore, there is no theoretical limit on how many constraints we can include (i.e. number of rows in $\mathscr{F}_{\mathbf{x}}$) – although in practice, of course, there is.

However, this is problematic mainly for two reasons. First of all, it makes the problem size grow. This increases memory requirements and execution time, and the numerical stability is worsen due to an increased condition number. This is especially clear from the fact that we want these observations to be noise-free, since the noise usually has a regularizing effect. Secondly, the constraints are only enforced point-wise, so a sample drawn from the posterior fulfills the constraint only in our chosen points. The obvious way of compensating for this is by increasing the number of points in which the constraints are observed – but that exacerbates the first problem. Clearly, the challenge grows quickly with the dimension of the inferred function.

Embedding the constraints in the covariance function removes these issues – it makes the enforcement continuous while the problem size is left unchanged. We will now address the question of how to design such a covariance function.

## 3.2 A new construction

We want to find a GP prior (2) such that any sample $\mathbf{f}(\mathbf{x})$ from that prior obeys the constraints (4). In turn, this leads to constraints on the mean and covariance functions (3) of that prior. However, instead of posing these constraints on the mean and covariance functions directly, we consider $\mathbf{f}(\mathbf{x})$ to be related to another function $\mathbf{g}(\mathbf{x})$ via some operator $\mathscr{G}_{\mathbf{x}}$

$$\mathbf{f}(\mathbf{x}) = \mathscr{G}_{\mathbf{x}}[\mathbf{g}]. \tag{5}$$

The constraints (4) then amounts to

$$\mathscr{F}_{\mathbf{x}}[\mathscr{G}_{\mathbf{x}}[\mathbf{g}]] = \mathbf{0}. \tag{6}$$

We would like this relation to be true for any function $\mathbf{g}(\mathbf{x})$. To do that, we will interpret $\mathscr{F}_{\mathbf{x}}$ and $\mathscr{G}_{\mathbf{x}}$ as matrices and use a similar procedure to that of solving systems of linear equations. Since $\mathscr{F}_{\mathbf{x}}$ and $\mathscr{G}_{\mathbf{x}}$ are linear operators, we can think of $\mathscr{F}_{\mathbf{x}}[\mathbf{f}]$ and $\mathscr{G}_{\mathbf{x}}[\mathbf{g}]$ as matrix-vector multiplications where $\mathscr{F}_{\mathbf{x}}[\mathbf{f}] = \mathscr{F}_{\mathbf{x}}\mathbf{f}$, with $(\mathscr{F}_{\mathbf{x}}\mathbf{f})_i = \sum_{j=1}^{K}(\mathscr{F}_{\mathbf{x}})_{ij}f_j$ where each element $(\mathscr{F}_{\mathbf{x}})_{ij}$ in the operator matrix $\mathscr{F}_{\mathbf{x}}$ is a scalar operator. With this notation, (6) can be written as

$$\mathscr{F}_{\mathbf{x}}\mathscr{G}_{\mathbf{x}} = \mathbf{0}. \tag{7}$$

This reformulation imposes constraints on the operator $\mathscr{G}_{\mathbf{x}}$ rather than on the GP prior for $\mathbf{f}(\mathbf{x})$ directly. We can now proceed by designing a GP prior for $\mathbf{g}(\mathbf{x})$ and transform it using the mapping (5). We further know that GPs are closed under linear operations. More specifically, if $\mathbf{g}(\mathbf{x})$ is modeled as a GP with mean $\boldsymbol{\mu}_{\mathbf{g}}(\mathbf{x})$ and covariance $K_{\mathbf{g}}(\mathbf{x}, \mathbf{x}')$, then $\mathbf{f}(\mathbf{x})$ is also a GP with

$$\mathbf{f}(\mathbf{x}) = \mathscr{G}_{\mathbf{x}}\mathbf{g} \sim \mathcal{GP}\left(\mathscr{G}_{\mathbf{x}}\,\boldsymbol{\mu}_{\mathbf{g}},\ \mathscr{G}_{\mathbf{x}}K_{\mathbf{g}}\mathscr{G}_{\mathbf{x}'}^{\mathsf{T}}\right). \tag{8}$$

We use $(\mathscr{G}_{\mathbf{x}}K_{\mathbf{g}}\mathscr{G}_{\mathbf{x}'}^{\mathsf{T}})_{ij}$ to denote that $(\mathscr{G}_{\mathbf{x}}K_{\mathbf{g}}\mathscr{G}_{\mathbf{x}'}^{\mathsf{T}})_{ij} = (\mathscr{G}_{\mathbf{x}})_{ik}(\mathscr{G}_{\mathbf{x}'})_{jl}(K_{\mathbf{g}})_{kl}$, where $\mathscr{G}_{\mathbf{x}}$ and $\mathscr{G}_{\mathbf{x}'}$ act on the first and second argument of $K_{\mathbf{g}}(\mathbf{x}, \mathbf{x}')$, respectively. See Section 2 in the Supplementary material for further details on linear operations on GPs.

The procedure to find the desired GP prior for $\mathbf{f}$ can now be divided into the following three steps

1. Find an operator $\mathscr{G}_{\mathbf{x}}$ that fulfills the condition (6).

2. Choose a mean and covariance function for $\mathbf{g}(\mathbf{x})$.
3. Find the mean and covariance functions for $\mathbf{f}(\mathbf{x})$ according to (8).

In addition to being resistant to the disadvantages of the approach described in Section 3.1, there are some additional strengths worth pointing out with this method. First of all, we have separated the task of encoding the constraints and encoding other desired properties of the kernel. The constraints are encoded in $\mathscr{F}_{\mathbf{x}}$ and the remaining properties are determined by the prior for $\mathbf{g}(\mathbf{x})$, such as smoothness assumptions. Hence, satisfying the constraints does not sacrifice any desired behavior of the target function.

Secondly, $K(\mathbf{x}, \mathbf{x}')$ is guaranteed to be a valid covariance function provided that $K_{\mathbf{g}}(\mathbf{x}, \mathbf{x}')$ is, since GPs are closed under linear functional transformations. From (8), it is clear that each column of $K$ must fulfill all constraints encoded in $\mathscr{F}_{\mathbf{x}}$. Possibly $K$ could be constructed only with this knowledge, assuming a general form and solving the resulting equation system. However, a solution may not just be hard to find, but one must also make sure that it is indeed a valid covariance function.

Furthermore, this approach provides a simple and straightforward way of constructing the covariance function even if the constraints have a complicated form. It makes no difference if the linear operators relate the components of the target function explicitly or implicitly – the procedure remains the same.

### 3.3 Illustrating example

We will now illustrate the method using the example (1) introduced already in the introduction. Consider a function $\mathbf{f}(\mathbf{x}) : \mathbb{R}^2 \mapsto \mathbb{R}^2$ satisfying $\frac{\partial f_1}{\partial x_1} + \frac{\partial f_2}{\partial x_2} = 0$, where $\mathbf{x} = [x_1,\ x_2]^{\mathsf{T}}$ and $\mathbf{f}(\mathbf{x}) = [f_1(\mathbf{x}),\ f_2(\mathbf{x})]^{\mathsf{T}}$. This equation describes all two-dimensional divergence-free vector fields. The constraint can be written as a linear constraint on the form (4) where $\mathscr{F}_{\mathbf{x}} = [\frac{\partial}{\partial x_1}\ \frac{\partial}{\partial x_2}]$ and $\mathbf{f}(\mathbf{x}) = [f_1(\mathbf{x})\ f_2(\mathbf{x})]^{\mathsf{T}}$. Modeling this function with a GP and building the covariance structure as described above, we first need to find the transformation $\mathscr{G}_{\mathbf{x}}$ such that (7) is fulfilled. For example, we could pick

$$\mathscr{G}_{\mathbf{x}} = \left[ -\frac{\partial}{\partial x_2}\ \ \frac{\partial}{\partial x_1} \right]^{\mathsf{T}}. \tag{9}$$

If the underlying function $g(\mathbf{x}) : \mathbb{R}^2 \mapsto \mathbb{R}$ is given by $g(\mathbf{x}) \sim \mathcal{GP}\big(0, k_g(\mathbf{x}, \mathbf{x}')\big)$, then we can make use of (8) to obtain $\mathbf{f}(\mathbf{x}) \sim \mathcal{GP}\big(\mathbf{0}, K(\mathbf{x}, \mathbf{x}')\big)$ where

$$K(\mathbf{x}, \mathbf{x}') = \mathscr{G}_{\mathbf{x}} k_g(\mathbf{x}, \mathbf{x}') \mathscr{G}_{\mathbf{x}}^{\mathsf{T}} = \begin{bmatrix} \frac{\partial^2}{\partial x_2 x_2'} & -\frac{\partial^2}{\partial x_2 x_1'} \\ -\frac{\partial^2}{\partial x_1 x_2'} & \frac{\partial^2}{\partial x_1 x_1'} \end{bmatrix} k_g(\mathbf{x}, \mathbf{x}').$$

Using a covariance function with the following structure, we know that the constraint will be fulfilled by any function generated from the corresponding GP.

## 4 Finding the operator $\mathscr{G}_{\mathbf{x}}$

In a general setting it might be hard to find an operator $\mathscr{G}_{\mathbf{x}}$ that fulfills the constraint (7). Ultimately, we want an algorithm that can construct $\mathscr{G}_{\mathbf{x}}$ from a given $\mathscr{F}_{\mathbf{x}}$. In more formal terms, the function $\mathscr{G}_{\mathbf{x}} \mathbf{g}$ forms the nullspace of $\mathscr{F}_{\mathbf{x}}$. The concept of nullspaces for linear operators is well-established [11], and does in many ways relate to real-number linear algebra.

However, an important difference is illustrated by considering a one-dimensional function $f(x)$ subject to the constraint $\mathscr{F}_x f = 0$ where $\mathscr{F}_x = \frac{\partial}{\partial x}$. The solution to this differential equation can not be expressed in terms of an arbitrary underlying function, but it requires $f(x)$ to be constant. Hence, the nullspace of $\frac{\partial}{\partial x}$ consists of the set of horizontal lines. Compare this with the real number equation $ab = 0, a \neq 0$, which is true only if $b = 0$. Since the nullspace differs between operators, we must be careful when discussing the properties of $\mathscr{F}_{\mathbf{x}}$ and $\mathscr{G}_{\mathbf{x}}$ based on knowledge from real-number algebra.

Let us denote the rows in $\mathscr{F}_{\mathbf{x}}$ as $\boldsymbol{\ell}_1^{\mathsf{T}}, \ldots, \boldsymbol{\ell}_L^{\mathsf{T}}$. We now want to find all solutions $\boldsymbol{g}$ such that

$$\mathscr{F}_{\mathbf{x}} \boldsymbol{g} = \mathbf{0} \quad \Rightarrow \quad \boldsymbol{\ell}_i^{\mathsf{T}} \boldsymbol{g} = 0, \quad \forall \quad i = 1, \ldots, L. \tag{10}$$

The solutions $\boldsymbol{g}_1, \ldots, \boldsymbol{g}_P$ to (10) will then be the columns of $\mathscr{G}_{\mathbf{x}}$. Each row vector $\boldsymbol{\ell}_j$ can be written as $\boldsymbol{\ell}_i = \Phi_i \boldsymbol{\xi}^{\boldsymbol{\ell}}$ where $\Phi_i \in \mathbb{R}^{K \times M_\ell}$ and $\boldsymbol{\xi}^{\boldsymbol{\ell}} = [\xi_1, \ldots, \xi_{M_\ell}]^{\mathsf{T}}$ is a vector of $M_\ell$ scalar operators

---

**Algorithm 1** Constructing $\mathcal{G}_{\mathbf{x}}$

---

    **Input:** Operator matrix $\mathcal{F}_{\mathbf{x}}$
    **Output:** Operator matrix $\mathcal{G}_{\mathbf{x}}$ where $\mathcal{F}_{\mathbf{x}}\mathcal{G}_{\mathbf{x}} = \mathbf{0}$
    **Step 1:** Make an ansatz $\boldsymbol{g} = \Gamma \boldsymbol{\xi}^{\boldsymbol{g}}$ for the columns in $\mathcal{G}_{\mathbf{x}}$.
    **Step 2:** Expand $\mathcal{F}_{\mathbf{x}}\Gamma \boldsymbol{\xi}^{\boldsymbol{g}}$ and collect terms.
    **Step 3:** Construct $A \cdot \text{vec}(\Gamma) = \mathbf{0}$ and find the vectors $\Gamma_1 \ldots \Gamma_P$ spanning its nullspace.
    **Step 4:** If $P = 0$, go back to **Step 1** and make a new ansatz, i.e. extend the set of operators.
    **Step 5:** Construct $\mathcal{G}_{\mathbf{x}} = [\Gamma_1 \boldsymbol{\xi}^{\boldsymbol{g}}, \ldots, \Gamma_P \boldsymbol{\xi}^{\boldsymbol{g}}]$.

---

included in $\mathcal{F}_{\mathbf{x}}$. We now assume that $\boldsymbol{g}$ also can be written in a similar form $\boldsymbol{g} = \Gamma \boldsymbol{\xi}^{\boldsymbol{g}}$ where $\Gamma \in \mathbb{R}^{K \times M_{\boldsymbol{g}}}$ and $\boldsymbol{\xi}^{\boldsymbol{g}} = [\xi_1, \ldots, \xi_{M_{\boldsymbol{g}}}]^{\mathsf{T}}$ is a vector of $M_{\boldsymbol{g}}$ scalar operators. One may make the assumption that the same set of operators that are used to describe $\boldsymbol{f}_i$ also can be used to describe $\boldsymbol{g}$, i.e., $\boldsymbol{\xi}^{\boldsymbol{g}} = \boldsymbol{\xi}^{\boldsymbol{f}}$. However, this assumption might need to be relaxed. The constraints (10) can then be written as

$$(\boldsymbol{\xi}^{\boldsymbol{f}})^{\mathsf{T}} \Phi_i \Gamma \boldsymbol{\xi}^{\boldsymbol{g}} = 0, \qquad \forall \quad i = 1, \ldots, L. \tag{11}$$

We perform the multiplication and collect the terms in $\boldsymbol{\xi}^{\boldsymbol{f}}$ and $\boldsymbol{\xi}^{\boldsymbol{g}}$. The condition (11) then results in conditions on the parameters in $\Gamma$ resulting a in a homogeneous system of linear equations

$$A \cdot \text{vec}(\Gamma) = \mathbf{0}. \tag{12}$$

The vectors $\text{vec}(\Gamma_1), \ldots, \text{vec}(\Gamma_P)$ spanning the nullspace of $A$ in (12) are then used to compute the columns in $\mathcal{G}_{\mathbf{x}} = [\boldsymbol{g}_1, \ldots \boldsymbol{g}_P]$ where $\boldsymbol{g}_p = \Gamma_p \boldsymbol{\xi}^{\boldsymbol{g}}$. If it turns out that the nullspace of $A$ is empty, one should start over with a new ansatz and extend the set of operators in $\boldsymbol{\xi}^{\boldsymbol{g}}$.

The outline of the procedure as described above is summarized in Algorithm 1. The algorithm is based upon a parametric ansatz rather than directly upon the theory for linear operators. Not only is it more intuitive, but it does also remove any conceptual challenges that theory may provide. A problem with this is that one may have to iterate before having found the appropriate set of operators in $\mathcal{G}_{\mathbf{x}}$. It might be of interest to examine possible alternatives to this algorithm that does not use a parametric approach. Let us now illustrate the method with an example.

## 4.1 Divergence-free example revisited

Let us return to the example discussed in Section 3.3, and show how the solution found by visual inspection also can be found with the algorithm described above. Since $\mathcal{F}_{\mathbf{x}}$ only contains first-order derivative operators, we assume that a column in $\mathcal{G}_{\mathbf{x}}$ does so as well. Hence, let us propose the following ansatz (step 1)

$$\boldsymbol{g} = \begin{bmatrix} \gamma_{11} & \gamma_{12} \\ \gamma_{21} & \gamma_{22} \end{bmatrix} \begin{bmatrix} \frac{\partial}{\partial x_1} \\ \frac{\partial}{\partial x_2} \end{bmatrix} = \Gamma \boldsymbol{\xi}^{\boldsymbol{g}}. \tag{13}$$

Applying the constraint, expanding and collecting terms (step 2) we find

$$\mathcal{F}_{\mathbf{x}} \Gamma \boldsymbol{\xi}^{\boldsymbol{g}} = \begin{bmatrix} \frac{\partial}{\partial x_1} & \frac{\partial}{\partial x_2} \end{bmatrix} \begin{bmatrix} \gamma_{11} & \gamma_{12} \\ \gamma_{21} & \gamma_{22} \end{bmatrix} \begin{bmatrix} \frac{\partial}{\partial x_1} \\ \frac{\partial}{\partial x_2} \end{bmatrix} = \gamma_{11} \frac{\partial^2}{\partial x_1^2} + (\gamma_{12} + \gamma_{21}) \frac{\partial^2}{\partial x_1 \partial x_2} + \gamma_{22} \frac{\partial^2}{\partial x_2^2}, \tag{14}$$

where we have used the fact that $\frac{\partial^2}{\partial x_i \partial x_j} = \frac{\partial^2}{\partial x_j \partial x_i}$ assuming continuous second derivatives. The expression (14) equals zero if

$$\begin{bmatrix} 1 & 0 & 0 & 0 \\ 0 & 1 & 1 & 0 \\ 0 & 0 & 0 & 1 \end{bmatrix} \begin{bmatrix} \gamma_{11} \\ \gamma_{12} \\ \gamma_{21} \\ \gamma_{22} \end{bmatrix} = A \cdot \text{vec}(\Gamma) = \mathbf{0}. \tag{15}$$

The nullspace is spanned by a single vector (step 3) $[\gamma_{11}\ \gamma_{12}\ \gamma_{21}\ \gamma_{22}]^{\mathsf{T}} = \lambda[0\ -1\ 1\ 0]^{\mathsf{T}}, \lambda \in \mathbb{R}$. Choosing $\lambda = 1$, we get $\mathcal{G}_{\mathbf{x}} = \left[-\frac{\partial}{\partial x_2} \quad \frac{\partial}{\partial x_1}\right]^{\mathsf{T}}$ (step 5), which is the same as in (9).

## 4.2 Generalization

Although there are no conceptual problems with the algorithm introduced above, the procedure of expanding and collecting terms appears a bit informal. In a general form, the algorithm is reformulated such that the operators are completely left out from the solution process. The drawback of this is a more cumbersome notation, and we have therefore limited the presentation to this simplified version. However, the general algorithm is found in the Supplementary material of this paper.

## 5 Experimental results

### 5.1 Simulated divergence-free function

Consider the example in Section 3.3. An example of a function fulfilling $\frac{\partial f_1}{\partial x_1} + \frac{\partial f_2}{\partial x_2} = 0$ is

$$
\begin{aligned}
f_1(x_1, x_2) &= e^{-ax_1x_2}\big(ax_1\sin(x_1x_2) - x_1\cos(x_1x_2)\big), \\
f_2(x_1, x_2) &= e^{-ax_1x_2}\big(x_2\cos(x_1x_2) - ax_2\sin(x_1x_2)\big),
\end{aligned}
\tag{16}
$$

where $a$ denotes a constant. We will now study how the regression of this function differs when using the covariance function found in Section 3.3 as compared to a diagonal covariance function $K(\mathbf{x}, \mathbf{x}') = k(\mathbf{x}, \mathbf{x}')I$. The measurements generated are corrupted with Gaussian noise such that $\mathbf{y}_k = \mathbf{f}(\mathbf{x}_k) + \mathbf{e}_k$, where $\mathbf{e}_k \sim \mathcal{N}(\mathbf{0}, \sigma^2 I)$. The squared exponential covariance function $k(\mathbf{x}, \mathbf{x}') = \sigma_f^2 \exp\left[-\frac{1}{2}l^{-2}\|\mathbf{x} - \mathbf{x}'\|^2\right]$ has been used for $k_g$ and $k$ with hyperparameters chosen by maximizing the marginal likelihood. We have used the value $a = 0.01$ in (16).

We have used 50 measurements randomly picked over the domain $[0\ 4] \times [0\ 4]$, generated with the noise level $\sigma = 10^{-4}$. The points for prediction corresponds to a discretization using 20 uniformly distributed points in each direction, and hence a total of $N_P = 20^2 = 400$. We have included the approach described is Section 3.1 for comparison. The number of artificial observations have been chosen as random subsets of the prediction points, up to and including the full set.

The comparison is made with regard to the root mean squared error $e_{\mathrm{rms}} = \sqrt{\frac{1}{N_P}\bar{\mathbf{f}}_\Delta^\top \bar{\mathbf{f}}_\Delta}$, where $\bar{\mathbf{f}}_\Delta = \hat{\bar{\mathbf{f}}} - \bar{\mathbf{f}}$ and $\bar{\mathbf{f}}$ is a concatenated vector storing the true function values in all prediction points and $\hat{\bar{\mathbf{f}}}$ denotes the reconstructed equivalent. To decrease the impact of randomness, each error value has been formed as an average over 50 reconstructions given different sets of measurements.

An example of the true field, measured values and reconstruction errors using the different methods is seen in Figure 2. The result from the experiment is seen in Figure 3a. Note that the error from the approach with artificial observations is decreasing as the number of observations is increased, but only to a certain point. Have in mind, however, that the Gram matrix is growing, making the problem larger and worse conditioned. The result from our approach is clearly better, while the problem size is kept small and numerical problems are therefore avoided.

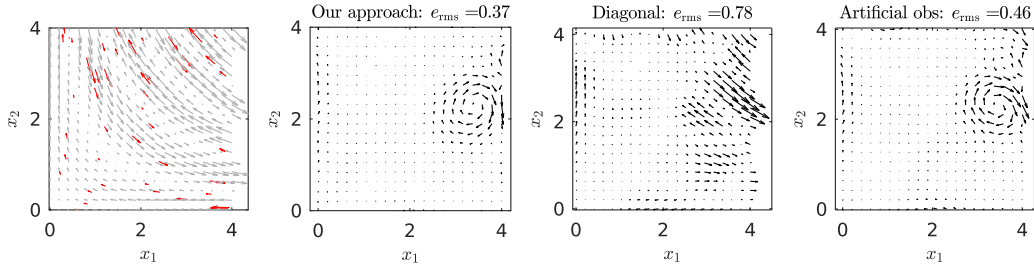

Figure 2: Left: Example of field plots illustrating the measurements (red arrows) and the true field (gray arrows). Remaining three plots: reconstructed fields subtracted from the true field. The artificial observations of the constraint have been made in the same points as the predictions are made.

### 5.2 Real data experiment

Magnetic fields can mathematically be considered as a vector field mapping a 3D position to a 3D magnetic field strength. Based on the magnetostatic equations, this can be modeled as a curl-free

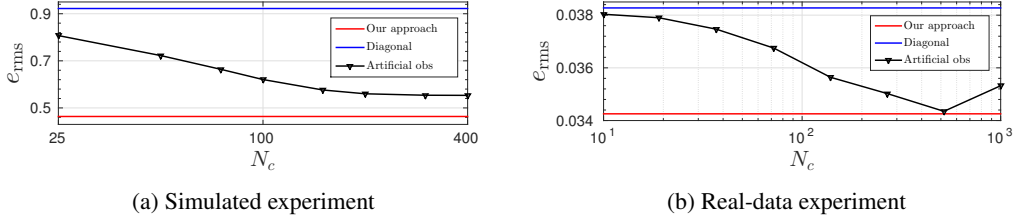

| (a) Simulated experiment | (b) Real-data experiment |

Figure 3: Accuracy of the different approaches as the number of artificial observations $N_c$ is increased.

vector field. Following Section 3.1 in the Supplementary material, our method can be used to encode the constraints in the following covariance function (which also has been presented elsewhere [22])

$$K_{\text{curl}}(\mathbf{x}, \mathbf{x}') = \sigma_f^2 e^{-\frac{\|\mathbf{x}-\mathbf{x}'\|^2}{2l^2}} \left( I_3 - \left( \frac{\mathbf{x} - \mathbf{x}'}{l} \right) \left( \frac{\mathbf{x} - \mathbf{x}'}{l} \right)^{\mathsf{T}} \right). \tag{17}$$

With a magnetic sensor and an optical positioning system, both position and magnetic field data have been collected in a magnetically distorted indoor environment, see the Supplementary material for details about the experimental details. In Figure 1 the predicted magnitude of the magnetic field over a two-dimensional domain for three different heights above the floor is displayed. The predictions have been made based on 500 measurements sampled from the trajectory given by the blue curve.

Similar to the simulated experiment in Section 5.1, we compare the predictions of the curl-free covariance function (17) with the diagonal covariance function and the diagonal covariance function using artificial observations. The results have been formed by averaging the error over 50 reconstructions. In each iteration, training data and test data were randomly selected from the data set collected in the experiment. 500 train data points and 1 000 test data points were used.

The result is seen in Figure 3b. We recognize the same behavior as we saw for the simulated experiment in Figure 3a. Note that the accuracy of the artificial observation approach gets very close to our approach for a large number of artificial observations. However, in the last step of increasing the artificial observations, the accuracy decreases. This is probably caused by the numerical errors that follows from an ill-conditioned Gram matrix.

## 6   Related work

Many problems in which GPs are used contain some kind of constraint that could be well exploited to improve the quality of the solution. Since there are a variety of ways in which constraints may appear and take form, there is also a variety of methods to deal with them. The treatment of inequality constraints in GP regression have been considered for instance in [1] and [5], based on local representations in a limited set of points. The paper [12] proposes a finite-dimensional GP-approximation to allow for inequality constraints in the entire domain.

It has been shown that linear constraints satisfied by the training data will be satisfied by the GP prediction as well [19]. The same paper shows how this result can be extended to quadratic forms through a parametric reformulation and minimization of the Frobenious norm, with application demonstrated for pose estimation. Another approach on capturing human body features is described in [18], where a face-shape model is included in the GP framework to imply anatomic correctness. A rigorous theoretical analysis of degeneracy and invariance properties of Gaussian random fields is found in [7], including application examples for one-dimensional GP problems. The concept of learning the covariance function with respect to algebraic invariances is explored in [9].

Although constraints in most situations are formulated on the outputs of the GP, there are also situations in which they are acting on the inputs. An example of this is given in [21], describing a method of benefit from ordering constraints on the input to reduce the negative impact of input noise.

Applications within medicine include gene-disease association through functional expectation constraints [10] and lung disease sub-type identification using a mixture of GPs and constraints encoded with Markov random fields [17]. Another way of viewing constraints is as modified prior distributions. By making use of the so-called multivariate generalized von Mises distribution, [13] ends up in a version of GP regression customized for circular variable problems. Other fields of interest include using GPs in approximately solving one-dimensional partial differential equations [8, 14, 15].

Generally speaking, the papers mentioned above consider problems in which the constraints are dealt with using some kind of external enforcement – that is, they are not explicitly incorporated into the model, but rely on approximations or finite representations. Therefore, the constraints may just be approximately satisfied and not necessarily in a continuous manner, which differs from the method proposed in this paper. Of course, comparisons can not be done directly between methods that have been developed for different kinds of constraints. The interest in this paper is multivariate problems where the constraints are linear combinations of the outputs that are known to equal zero.

For multivariate problems, constructing the covariance function is particularly challenging due to the correlation between the output components. We refer to [2] for a very useful review. The basic idea behind the so-called *separable kernels* is to separate the process of modeling the covariance function for each component and the process of modeling the correlation between them. The final covariance function is chosen for example according to some method of regularization. Another class of covariance functions is the *invariant kernels*. Here, the correlation is inherited from a known mathematical relation. The curl- and divergence free covariance functions are such examples where the structure follows directly from the underlying physics, and has been shown to improve the accuracy notably for regression problems [22]. Another example is the method proposed in [4], where the Taylor expansion is used to construct a covariance model given a known relationship between the outputs. A very useful property on linear transformations is given in [20], based on the GPs natural inheritance of features imposed by linear operators. This fact has for example been used in developing a method for monitoring infectious diseases [3].

The method proposed in this work is exploiting the transformation property to build a covariance function of the invariant kind for a multivariate GP. We show how this property can be exploited to incorporate knowledge of linear constraints into the covariance function. Moreover, we present an algorithm of constructing the required transformation. This way, the constraints are built into the prior and are guaranteed to be fulfilled in the entire domain.

# 7    Conclusion and future work

We have presented a method for designing the covariance function of a multivariate Gaussian process subject to known linear operator constraints on the target function. The method will by construction guarantee that any sample drawn from the resulting process will obey the constraints in all points. Numerical simulations show the benefits of this method as compared to alternative approaches. Furthermore, it has been demonstrated to improve the performance on real data as well.

As mentioned in Section 4, it would be desirable to describe the requirements on $\mathscr{G}_{\mathbf{x}}$ more rigorously. That might allow us to reformulate the construction algorithm for $\mathscr{G}_{\mathbf{x}}$ in a way that allows for a more straightforward approach as compared to the parametric ansatz that we have proposed. In particular, our method relies upon the requirement that the target function can be expressed in terms of an underlying *potential* function g. This leads to the intriguing and nontrivial question: Is it possible to mathematically guarantee the existence of such a potential? If the answer to this question is yes, the next question will of course be what it look like and how it relates to the target function.

Another possible topic of further research is the extension to constraints including *nonlinear* operators, which for example might rely upon a linearization in the domain of interest. Furthermore, it may be of potential interest to study the extension to a non-zero right-hand side of (4).

**Acknowledgements**

This research is financially supported by the Swedish Foundation for Strategic Research (SSF) via the project *ASSEMBLE* (Contract number: RIT 15-0012). The work is also supported by the Swedish Research Council (VR) via the project *Probabilistic modeling of dynamical systems* (Contract number: 621-2013-5524). We are grateful for the help and equipment provided by the UAS Technologies Lab, Artificial Intelligence and Integrated Computer Systems Division (AIICS) at the Department of Computer and Information Science (IDA), Linköping University, Sweden. The real data set used in this paper has been collected by some of the authors together with Manon Kok, Arno Solin, and Simo Särkkä. We thank them for allowing us to use this data. We also thank Manon Kok for supporting us with the data processing. Furthermore, we would like to thank Carl Rasmussen and Marc Deisenroth for fruitful discussions on constrained GPs.

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
