[Supplementary Material · supplementary_nips.pdf]

# Supplementary Material
# Linearly constrained Gaussian processes

**Carl Jidling**
Department of Information Technology
Uppsala University, Sweden
carl.jidling@it.uu.se

**Niklas Wahlström**
Department of Information Technology
Uppsala University, Sweden
niklas.wahlstrom@it.uu.se

**Adrian Wills**
School of Engineering
University of Newcastle, Australia
adrian.wills@newcastle.edu.au

**Thomas B. Schön**
Department of Information Technology
Uppsala University, Sweden
thomas.schon@it.uu.se

## 1 Linear operators

In this work we consider linear operators on functions. Such an operator transforms a function $\mathbf{f}(\mathbf{x})$ to another function $\mathbf{g}(\mathbf{z})$. We denote this according to

$$\mathbf{g}(\mathbf{z}) = \mathscr{F}_{\mathbf{z}}[\mathbf{f}(\mathbf{x})]. \tag{1}$$

This linear operator could be *differentiation* of a function. If $D = 1$ and $K = 1$ this will be defined as

$$g(z) = \mathscr{F}_z[f] = \left.\frac{\partial f(x)}{\partial x}\right|_{x=z} \tag{2a}$$

which slightly more informal also can be written as

$$g(x) = \mathscr{F}_x[f] = \frac{\partial f(x)}{\partial x}. \tag{2b}$$

Also *integration* of a scalar function $f(x)$ over an interval $[z_1, z_2]$ is a linear operator

$$g(\mathbf{z}) = \mathscr{F}_{\mathbf{z}}[f] = \int_{z_1}^{z_2} f(x)dx, \tag{3}$$

where $g(\mathbf{z})$ is a scalar-valued function with a two-dimensional input $\mathbf{z} = [z_1,\ z_2]^{\mathsf{T}}$. Note that in the two examples given above, the inputs of $f$ and $g$ will not be the same, not even of the same dimension!

Input wrapping is another way to construct new covariance functions from old ones [4, page 92]. It utilizes a nonlinear wrapping $\mathbf{x} = \mathbf{u}(\mathbf{z})$ of the input variables. This wrapping can also be considered as a linear operator, where

$$\mathbf{g}(\mathbf{z}) = \mathscr{F}_{\mathbf{z}}[\mathbf{f}] = \mathbf{f}(\mathbf{x})|_{\mathbf{x}=\mathbf{u}(\mathbf{z})}. \tag{4}$$

This operator also changes the function input and possibly also its dimension. Even though the wrapping itself might be nonlinear, the operator corresponding to this wrapping is in fact linear.

It is straightforward to show that all three operators presented above do fulfill the linearity condition.

## 2   Gaussian processes under linear operations

It is well-known that Gaussian distributions are closed under linear transformation. In similar manner, Gaussian processes are closed under linear operations [3, 4, 2, 1].

By applying the functional $\mathcal{F}_{\mathbf{x}}$ on both the mean function and the covariance function, the GP prior for $\mathcal{F}_{\mathbf{x}}$ is given by

$$\mathcal{F}_{\mathbf{x}}\mathbf{f} \sim \mathcal{GP}\left(\mathcal{F}_{\mathbf{x}}\,\boldsymbol{\mu},\ \mathrm{Cov}\left[\mathcal{F}_{\mathbf{x}}\mathbf{f}(\mathbf{x}),\ \mathcal{F}_{\mathbf{x}'}\mathbf{f}(\mathbf{x}')\right]\right). \tag{5}$$

The covariance becomes

$$\begin{aligned}
&\mathrm{Cov}\left[\mathcal{F}_{\mathbf{x}}\mathbf{f}(\mathbf{x}),\ \mathcal{F}_{\mathbf{x}'}\mathbf{f}(\mathbf{x}')\right] \\
&= \mathbb{E}\left[\left(\mathcal{F}_{\mathbf{x}}\mathbf{f}(\mathbf{x}) - \mathcal{F}_{\mathbf{x}}\boldsymbol{\mu}(\mathbf{x})\right)\left(\mathcal{F}_{\mathbf{x}'}\mathbf{f}(\mathbf{x}') - \mathcal{F}_{\mathbf{x}'}\boldsymbol{\mu}(\mathbf{x}')\right)^{\mathsf{T}}\right] \\
&= \mathcal{F}_{\mathbf{x}}\mathbb{E}\left[\left(\mathbf{f}(\mathbf{x}) - \boldsymbol{\mu}(\mathbf{x})\right)\left(\mathbf{f}(\mathbf{x}') - \boldsymbol{\mu}(\mathbf{x}')\right)^{\mathsf{T}}\right]\mathcal{F}_{\mathbf{x}'}^{\mathsf{T}} \\
&= \mathcal{F}_{\mathbf{x}}K\mathcal{F}_{\mathbf{x}'}^{\mathsf{T}},
\end{aligned} \tag{6}$$

where by the notation $(\mathcal{F}_{\mathbf{x}}K\mathcal{F}_{\mathbf{x}'}^{\mathsf{T}})_{ij}$ we mean that

$$(\mathcal{F}_{\mathbf{x}}K\mathcal{F}_{\mathbf{x}'}^{\mathsf{T}})_{ij} = (\mathcal{F}_{\mathbf{x}})_{ik}(\mathcal{F}_{\mathbf{x}'})_{jl}K_{kl}, \tag{7}$$

and where $(\mathcal{F}_{\mathbf{x}})_{ik}$ and $(\mathcal{F}_{\mathbf{x}'})_{jl}$ act on the first and second argument of $K_{kl}(\mathbf{x}, \mathbf{x}')$, respectively.

We should point out that some care must be taken when applying this procedure. For example, if we would like to consider the derivative of a function governed by a GP, we must make sure that this function is modeled in a way such that the derivative actually exists. This may sound obvious, yet important to remember since the set of standard covariance functions includes members that are not differentiable – among those we find $\mathrm{Matérn}_{1/2}$ [4].

## 3   Generalization of Section 4

In this supplementary material we will generalize the method described in the main paper on how to solve operator matrix equations on the form

$$\mathcal{F}\mathcal{G} = \mathbf{0},$$

where we want to find $\mathcal{G}$ given $\mathcal{F}$ [1]. If $\mathcal{F} \in \mathbb{R}^{m \times n}$ is a real valued matrix, $\mathcal{G}$ can easily be found by letting the columns in $\mathcal{G}$ span the nullspace of $\mathcal{F}$ (provided such a nullspace exist). However, if the elements of $\mathcal{F}$ are operators, the situation is more tricky. This supplementary material generalizes the parametric approach presented in Section 4 in the main paper for arbitrary operators of any order. The strategy is to study the vector space of homogeneous polynomials where the operators are interpreted as the variables of these polynomials.

In Section 3.1, we assume that both $\mathcal{F}$ and $\mathcal{G}$ consist of first order operators and in Section 3.2 we generalize this to allow for any order of the operators.

### 3.1   First order operator equation

Consider the matrix $\mathcal{F} \in \mathcal{P}_p^{m \times n}$, where $\mathcal{P}_p$ is a vector space of first order operators

$$\mathcal{P}_p = \{a_1 y_1 + \ldots a_p y_p \,|\, a_1, \ldots, a_p \in \mathbb{R}\}, \tag{8}$$

where $y_1, \ldots, y_p$ is the basis in that vector space. The basis components $y_k$ can for example represent derivative operators $y_k = \frac{\partial}{\partial x_k}$. We want to find the vectors $\boldsymbol{g} \in \mathcal{P}_p^n$ such that $\mathcal{F}\boldsymbol{g} = \mathbf{0}$ is fulfilled. We can write $\mathcal{F} \in \mathcal{P}_p^{m \times n}$ and $\boldsymbol{g} \in \mathcal{P}_p^n$ as

$$\mathcal{F}_{ij} = \sum_{k=1}^{p} \phi_{ijk} y_k, \qquad \phi_{ijk} = \{\Phi\}_{ijk} \in \mathbb{R}, \tag{9a}$$

$$\boldsymbol{g}_j = \sum_{k=1}^{p} \gamma_{jk} y_k, \qquad \gamma_{jk} = \{\Gamma\}_{jk} \in \mathbb{R}, \tag{9b}$$

where $\Phi \in \mathbb{R}^{m \times n \times p}$ and $\Gamma \in \mathbb{R}^{n \times p}$. This gives

$$\mathscr{F}\boldsymbol{g} = \mathbf{0} \Leftrightarrow \sum_{j=1}^{n} \sum_{k=1}^{p} \sum_{l=1}^{p} \phi_{ijk} y_k \gamma_{jl} y_l = 0 \quad \forall\ i = 1:m. \tag{10}$$

For each $i$, we have a quadratic form

$$\mathbf{y}^\mathsf{T} \Phi_i \Gamma \mathbf{y} = 0, \tag{11}$$

where $\Phi_i \in \mathbb{R}^{p \times n}$ with $\{\Phi_i\}_{kj} = \phi_{ijk}$ and $\Gamma \in \mathbb{R}^{n \times p}$ with $\{\Gamma\}_{jk} = \gamma_{jk}$.

The quadratic form is equal to zero for all $\mathbf{y}$ if and only if

$$\Phi_i \Gamma + \Gamma^\mathsf{T} \Phi_i^\mathsf{T} = \mathbf{0} \quad \forall \quad i = 1:m. \tag{12}$$

### Example 1 (divergence free vector field)

We consider the following vector of operators $\mathscr{F} \in \mathcal{P}_3^{1 \times 3}$

$$\mathscr{F} = \nabla_\mathbf{x} = \left[ \frac{\partial}{\partial x_1},\ \frac{\partial}{\partial x_2},\ \frac{\partial}{\partial x_3} \right], \tag{13}$$

where

$$\mathscr{F}_{ij} = \sum_{k=1}^{3} \phi_{ijk} y_k, \quad \forall \quad i = 1, \quad j = 1, 2, 3, \tag{14}$$

where $y_k = \frac{\partial}{\partial x_k}$. Following the notation introduced above, for this particular operator matrix we have

$$\Phi_1 = \begin{bmatrix} 1 & 0 & 0 \\ 0 & 1 & 0 \\ 0 & 0 & 1 \end{bmatrix}. \tag{15}$$

We now want of find a vector $\boldsymbol{g} \in \mathcal{P}^3$ that fulfills $\mathscr{F}\boldsymbol{g} = \mathbf{0}$ for all $\mathbf{y}$. We assume that this operator vector is in $\boldsymbol{g} \in \mathcal{P}_3^3$ and can be written

$$\boldsymbol{g}_j = \sum_{k=1}^{3} \gamma_{jk} y_k \quad j = 1, 2, 3, \tag{16}$$

where $\Gamma \in \mathbb{R}^{3 \times 3}$ is unknown. Now we have that

$$\Phi_1 \Gamma + \Gamma^\mathsf{T} \Phi_1^\mathsf{T} = \mathbf{0} \tag{17a}$$

$$\Rightarrow \begin{bmatrix} \gamma_{11} & \gamma_{12} - \gamma_{21} & \gamma_{13} - \gamma_{31} \\ \gamma_{21} - \gamma_{12} & \gamma_{22} & \gamma_{23} - \gamma_{32} \\ \gamma_{31} - \gamma_{13} & \gamma_{32} - \gamma_{23} & \gamma_{33} \end{bmatrix} = \mathbf{0}, \tag{17b}$$

which in turn gives

$$\gamma_{11} = 0, \qquad \gamma_{12} + \gamma_{21} = 0, \tag{18a}$$
$$\gamma_{22} = 0, \qquad \gamma_{13} + \gamma_{31} = 0, \tag{18b}$$
$$\gamma_{33} = 0, \qquad \gamma_{23} + \gamma_{32} = 0. \tag{18c}$$

The nullspace of (17a) is then spanned by

$$\Gamma = \lambda_1 \begin{bmatrix} 0 & 0 & 0 \\ 0 & 0 & 1 \\ 0 & \text{-}1 & 0 \end{bmatrix} + \lambda_2 \begin{bmatrix} 0 & 0 & \text{-}1 \\ 0 & 0 & 0 \\ 1 & 0 & 0 \end{bmatrix} + \lambda_3 \begin{bmatrix} 0 & 1 & 0 \\ \text{-}1 & 0 & 0 \\ 0 & 0 & 0 \end{bmatrix},$$

which gives

$$\boldsymbol{g} = \lambda_1 \begin{bmatrix} 0 \\ \frac{\partial}{\partial x_3} \\ \text{-}\frac{\partial}{\partial x_2} \end{bmatrix} + \lambda_2 \begin{bmatrix} \text{-}\frac{\partial}{\partial x_3} \\ 0 \\ \frac{\partial}{\partial x_1} \end{bmatrix} + \lambda_3 \begin{bmatrix} \frac{\partial}{\partial x_2} \\ \text{-}\frac{\partial}{\partial x_1} \\ 0 \end{bmatrix}, \lambda_1, \lambda_2, \lambda_3 \in \mathbb{R}.$$

**Example 2 (curl free vector field)**

We consider the following vector of operators $\mathscr{F} \in \mathcal{P}_3^{3 \times 3}$

$$\mathscr{F} = \begin{bmatrix} 0 & \frac{\partial}{\partial x_3} & -\frac{\partial}{\partial x_2} \\ -\frac{\partial}{\partial x_3} & 0 & \frac{\partial}{\partial x_1} \\ \frac{\partial}{\partial x_2} & -\frac{\partial}{\partial x_1} & 0 \end{bmatrix}, \tag{19}$$

where

$$\mathscr{F}_{ij} = \sum_{k=1}^{3} \phi_{ijk} y_k, \quad \forall \, i = 1:3, \quad j = 1:3, \tag{20}$$

where $y_k = \frac{\partial}{\partial x_k}$. For this particular operator matrix we have

$$\Phi_1 = \begin{bmatrix} 0 & 0 & 0 \\ 0 & 0 & \text{-}1 \\ 0 & 1 & 0 \end{bmatrix}, \quad \Phi_2 = \begin{bmatrix} 0 & 0 & 1 \\ 0 & 0 & 0 \\ \text{-}1 & 0 & 0 \end{bmatrix}, \quad \Phi_3 = \begin{bmatrix} 0 & \text{-}1 & 0 \\ 1 & 0 & 0 \\ 0 & 0 & 0 \end{bmatrix}.$$

We now want to find a vector $\boldsymbol{g} \in \mathcal{P}^3$ which fulfills $\mathscr{F}\boldsymbol{g} = 0$ for all $\mathbf{y}$. We assume that this operator vector is in $\boldsymbol{g} \in \mathcal{P}_3^3$ and can be written

$$\boldsymbol{g}_j = \sum_{k=1}^{3} \gamma_{jk} y_k \quad j = 1, 2, 3, \tag{21}$$

where $\Gamma \in \mathbb{R}^{3 \times 3}$ is unknown. Now we have that

$$\Phi_1 \Gamma + \Gamma^\mathsf{T} \Phi_1^\mathsf{T} = \mathbf{0} \Rightarrow \begin{bmatrix} 0 & \text{-}\gamma_{31} & \gamma_{21} \\ \text{-}\gamma_{31} & \text{-}2\gamma_{32} & \gamma_{22}\text{-}\gamma_{33} \\ \gamma_{21} & \gamma_{22}\text{-}\gamma_{33} & 2\gamma_{23} \end{bmatrix} = \mathbf{0},$$

$$\Phi_2 \Gamma + \Gamma^\mathsf{T} \Phi_2^\mathsf{T} = \mathbf{0} \Rightarrow \begin{bmatrix} 2\gamma_{31} & \gamma_{32} & \gamma_{33}\text{-}\gamma_{11} \\ \gamma_{32} & 0 & \text{-}\gamma_{12} \\ \gamma_{33}\text{-}\gamma_{11} & \text{-}\gamma_{12} & \text{-}2\gamma_{13} \end{bmatrix} = \mathbf{0},$$

$$\Phi_3 \Gamma + \Gamma^\mathsf{T} \Phi_3^\mathsf{T} = \mathbf{0} \Rightarrow \begin{bmatrix} 2\gamma_{21} & \gamma_{22}\text{-}\gamma_{11} & \gamma_{23} \\ \gamma_{22}\text{-}\gamma_{11} & \text{-}2\gamma_{12} & \text{-}\gamma_{13} \\ \gamma_{23} & \text{-}\gamma_{13} & 0 \end{bmatrix} = \mathbf{0},$$

which in turn gives

$$\gamma_{22} - \gamma_{33} = 0, \qquad \gamma_{23} = 0, \qquad \gamma_{32} = 0, \tag{22a}$$
$$\gamma_{33} - \gamma_{11} = 0, \qquad \gamma_{13} = 0, \qquad \gamma_{31} = 0, \tag{22b}$$
$$\gamma_{22} - \gamma_{11} = 0, \qquad \gamma_{12} = 0, \qquad \gamma_{21} = 0. \tag{22c}$$

The nullspace of (22a) is then spanned by the single base vector

$$\Gamma = \lambda_1 \begin{bmatrix} 1 & 0 & 0 \\ 0 & 1 & 0 \\ 0 & 0 & 1 \end{bmatrix}, \quad \lambda_1 \in \mathbb{R}, \tag{23}$$

which gives

$$\boldsymbol{g} = \lambda_1 \begin{bmatrix} \frac{\partial}{\partial x_1} \\ \frac{\partial}{\partial x_2} \\ \frac{\partial}{\partial x_3} \end{bmatrix}, \quad \lambda_1 \in \mathbb{R}. \tag{24}$$

The final covariance function becomes

$$K(\mathbf{x}, \mathbf{x}') = \begin{bmatrix} \frac{\partial^2}{\partial x_1 \partial x_1'} & \frac{\partial^2}{\partial x_1 \partial x_2'} & \frac{\partial^2}{\partial x_1 \partial x_3'} \\ \frac{\partial^2}{\partial x_2 \partial x_1'} & \frac{\partial^2}{\partial x_2 \partial x_2'} & \frac{\partial^2}{\partial x_2 \partial x_3'} \\ \frac{\partial^2}{\partial x_3 \partial x_1'} & \frac{\partial^2}{\partial x_3 \partial x_2'} & \frac{\partial^2}{\partial x_3 \partial x_3'} \end{bmatrix} k_g(\mathbf{x}, \mathbf{x}'). \tag{25}$$

Figure 1: Three snapshots from the measurement collection. The senor platform was moved around by hand during approximately three minutes.

If we use the squared exponential covariance function

$$k_g(\mathbf{x}, \mathbf{x}') = \sigma_f^2 e^{-\frac{\|\mathbf{x}-\mathbf{x}'\|^2}{2l^2}} \tag{26}$$

we get

$$K(\mathbf{x}, \mathbf{x}') = \frac{\sigma_f^2}{l^2} e^{-\frac{\|\mathbf{x}-\mathbf{x}'\|^2}{2l^2}} \left( I_3 - \left( \frac{\mathbf{x}-\mathbf{x}'}{l} \right) \left( \frac{\mathbf{x}-\mathbf{x}'}{l} \right)^\mathsf{T} \right). \tag{27}$$

This covariance function is used in the real data experiment in Section 5.2 of the main paper. Note, that the version in the paper does not use $l^2$ in the denominator (which we also would get here if we would multiply (24) with $l^2$, still providing the same constraints).

## 3.2 Higher order operator equation

Now, consider the matrix $\mathscr{F} \in \mathcal{P}_{p,q}^{m \times n}$, where $\mathcal{P}_{p,q}$ is a vector space of all homogeneous polynomials of degree $q$ in $p$ variables

$$\mathcal{P}_{p,q} = \left\{ \sum_{k_1}^p \cdots \sum_{k_q}^p a_{k_1,\dots,k_q} y_{k_1} \cdots y_{k_q} \Gamma ig | a_{k_1,\dots,k_q} \in \mathbb{R} \right\},$$

where the nominals $y_{k_1} \cdots y_{k_q}$ constitute the basis of that vector space. The components $y_k$ can for example represent derivative operators $y_k = \frac{\partial}{\partial x_k}$ and $\mathcal{P}_{p,q}$ then contain all $q$th order derivatives of $x_1 \dots x_q$. We want to find the vectors $\boldsymbol{g} \in \mathcal{P}_{p,q_g}^n$ such that $\mathscr{F}\boldsymbol{g} = \mathbf{0}$ is fulfilled. We can write $\mathscr{F} \in \mathcal{P}_{p,q}^{m \times n}$ and $\boldsymbol{g} \in \mathcal{P}_{p,q_g}^n$ as

$$\mathscr{F}_{ij} = \sum_{k_1}^p \cdots \sum_{k_q}^p \phi_{i,j,k_1,\dots,k_q} y_{k_1} \cdots y_{k_q}, \tag{28a}$$

$$\boldsymbol{g}_j = \sum_{k_1}^p \cdots \sum_{k_q}^p \gamma_{j,k_1,\dots,k_{q_g}} y_{k_1} \cdots y_{k_{q_g}}, \tag{28b}$$

where $\Phi \in \mathbb{R}^{m \times n \times p^{\times q}}$ and $\mathbf{b} \in \mathbb{R}^{n \times p^{\times q}}$ (here $p^{\times q}$ denotes $\underbrace{p \times \cdots \times p}_{q \text{ times}}$). This gives

$$\mathscr{F}\boldsymbol{g} = \mathbf{0} \Leftrightarrow \sum_j^n \sum_{k_1}^p \cdots \sum_{k_q}^p \sum_{l_1}^p \cdots \sum_{l_q}^p \Big\{$$

$$\phi_{ijk_1\dots k_q} y_{k_1} \cdots y_{k_q} \gamma_{jl_1\dots l_{q_g}} y_{l_1} \cdots y_{l_{q_g}} \Big\} = 0 \quad \forall\, i = 1:m.$$

For each $i$, this is an algebraic form of order $q + q_g$

$$\sum_j^n \sum_{k_1\dots k_q,\, l_1\dots l_q \in \{d_1\dots d_{q+q_g}\}} \phi_{ijd_1\dots d_q} \gamma_{jd_{q+1}\dots d_{q+q_g}} = 0$$

$$\forall \quad i = 1:m, \quad k_1 = 1:p, \quad \dots, \quad k_q = 1:p,$$
$$l_1 = 1:p, \quad \dots, \quad l_q = 1:p,$$

where the second sum sums over all permutations of $k_1 \dots k_q, l_1 \dots l_q$.

# 4 Real data experiment description

This section contains more details about the real data experiment described in Section 5.2.

## 4.1 Experiment setup

To collect the measurements we made use of a wooden platform, see Figure 2. The platform was equipped with a Trivisio Colibri wireless IMU (TRIVISIO Prototyping GmbH, http://www.trivisio.com/), sampled at 100 Hz. The sensor includes both an accelerometer, a gyroscope, and a magnetometer. For additional validation a Google Nexus 5 smartphone was also mounted on the platform even tough its data was never used in this experiment.

Figure 2: Platform with magnetic sensors. The sensor to the left is the Trivisio sensor, whose magnetometer data we used during the experiment. The platform was also equipped with multiple markers visible to an optical reference system (Vicon).

On the platform, five markers were mounted. An optical reference system (Vicon) with several cameras mounted in the ceiling measured the 3D position of each marker, and hence also the position and the orientation of the platform relative to its predefined origin.

## 4.2 Experiment execution

The sensor platform was moved around by hand up and down in a volume of $4 \times 4 \times 2$ meters, see Figure 1. During the experiment, measurements were collected from the sensors on the platform as well from the optical reference system. The data from the different sensors were collected asynchronously. The experiment lasted for 187 seconds.

## 4.3 Pre-processing of data

The position and orientation data from the optical reference system was synchronized with the data from the Trivisio sensor. The synchronization was performed based on correlation analysis of the angular velocities measured by both systems.

The position in global coordinates of the Trivisio sensor was computed based on the position data, the orientation data, and the displacement of the Trivisio sensor relative to the predefined origin of the platform.

The magnetometer data from the Trivisio sensor was rotated from sensor-fixed coordinates to global coordinates using the orientation data from the optical reference system. These rotated measurements describe the magnetic field in global coordinates at the sensor positions computed above. In Section 5.2 of the main paper, these position data and magnetic field data are considered as input and output data, respectively.

## Footnotes

[1]In this supplementary material, the argument $\mathbf{x}$ is omitted for simplified notation