[Reviews · NeurIPS 2017]

Reviewer 1



%%% UPDATE: Thank you for your response, which has been read %%% This paper addresses an interesting topic - how to encode symmetries or invariances into kernels. The principal application area is in Bayesian nonparametric regression, where constraints may be presented as prior information but the form of the regression function is not fully specified. An important example of such a constraint is when the field being modelled is known to be of the divergence kind, so that a certain physical quantity is conserved. Related work in this area has included both ANOVA kernels and Stein's method-based kernel, which are each constrained to have vanishing integrals. The present work differs, in that it attempts to consider more general forms of constraint. This effort is commendable, as generic kernels continue to be widely used in machine learning applications without strong physical justification. I think the proposed method is not general and there is no theory to suggest that it will work in anything other than some special cases, but nevertheless it is interesting and could reasonably be in the proceedings for NIPS. - The authors ought to discuss their work in relation to "Learning with Algebraic Invariances, and the Invariant Kernel Trick" by Franz J. Király, Andreas Ziehe, Klaus-Robert Müller. - On page 4 it is incorrectly claimed that the prior for f will "inherit" the properties of the prior for g. This is of course not true - if g has a prior GP(0,k) where k generates a Sobolev space of order b and if \mathcal{F} is a differential operator of order c, then the prior on g will be something like a Sobolev space of order b - c. So the smoothness properties of f and g differ. - On page 5, it reads as though the equation "f_i = \Phi_i \xi f" is without loss of generality, but it is of course an ansatz. This could be better emphasised by clearly labelling this (and the equation "g = \Gamma \xi^g") as an ansatz. - The limited generality of the method could be better acknowledged. Perhaps an example where the method fails could be included.

Reviewer 2



Summary of the Paper: This paper describes a mechanism to incorporate linear operator constraints in the framework of Gaussian process regression. For this, the mean function and the covariance function of the Gaussian processes are changed. The aim of this transformation is to guarantee that samples from the GP posterior distribution satisfy the constraints indicated. These constraints are typically in the form of partial derivatives, although any linear operator can be considered in practice, e.g., integration too. Traditional methods incorporated these constraints by introducing additional observations. This has the limitation that is more expensive and restricted to the observations made. The framework proposed is evaluated in a synthetic problem and in a real problem, showing benefits with respect to the data augmentation strategy. Detailed comments: Quality: I think the quality of the paper is good in general. It is a very well written paper. Furthermore, all the key points are carefully described. It also has a strong related work section. The weakest point is however, the section on experiments in which only a synthetic dataset and a real dataset is considered. Clarity: The clarity of the paper is high. Originality: As far as I know the paper seems original. There are some related methods in the literature. However, they simply augment the observed data with virtual instances that have the goal of guaranteeing the constraints imposed. Significance: I think the problem addressed by the authors is relevant and important for the machine learning community. However, I have the feeling that the authors have not success in noting this. The examples used by the authors are a bit simple. For example they only consider a single real example and only the linear operator of derivatives. I have the feeling that this paper may have potential applications in probabilistic numeric methods, in which often a GP is used. Summing up, I think that this is a good paper. However, the weak experimental section questions its significance. Furthermore, I find difficult to find practical applications within the machine learning community. The authors should have made a bigger effort on showing this. I would consider it hence a borderline paper.

Reviewer 3



The authors present a novel method for inference in Gaussian processes subject to linear equality constraints. In contrast to previous approaches, which used techniques such as data augmentation with artificial observations, the proposed method incorporates the linear constraints directly into the GP kernel such that all draws from the GP satisfy the constraints. This is an elegant solution for linearly constrained GP regression problems. The principal drawback of this approach appears to be the non-trivial task of finding an operator G_x that spans the nullspace of F_x. Algorithm 1 suggests an iterative approach to constructing such an operator, but little guidance is given for the crucial step of selecting a set of scalar operators (\xi_g). For the running example, the differential operators are natural guesses given the form of F_x, but how might this be done more generally? Moreover, is it guaranteed that such an operator exists? Minor comments: - Section 3.1 seems out of place. It would naturally fall under "Related Work," but that is currently deferred to the end. Personally, I think Related Work should come before Section 3. - The discussion of "interpreting F_x and G_x as matrices" and "thinking of F_x[f] as matrix-vector multiplications" is a bit informal and could be made more rigorous. As written, the reader is left wondering if/when this is warranted.